# Reconstructing higher-order interactions in coupled dynamical systems

Federico Malizia[1,2], Alessandra Corso[3], Lucia Valentina Gambuzza[3], Giovanni Russo[4], Vito Latora[1,5,6,7] & Mattia Frasca [3] ✉

Higher-order interactions play a key role for the operation and function of a complex system. However, how to identify them is still an open problem. Here, we propose a method to fully reconstruct the structural connectivity of a system of coupled dynamical units, identifying both pairwise and higher-order interactions from the system time evolution. Our method works for any dynamics, and allows the reconstruction of both hypergraphs and simplicial complexes, either undirected or directed, unweighted or weighted. With two concrete applications, we show how the method can help understanding the complexity of bacterial systems, or the microscopic mechanisms of interaction underlying coupled chaotic oscillators.

Higher-order interactions are present in ecosystems, in which the way two species interact can be influenced by a third species[1], in social systems, where interactions in groups of three or more individuals naturally occur[2], in the brain cortex[3], and in many other complex systems[4]. Recent studies based on mathematical tools such as simplicial complexes[5,6] and hypergraphs[7] have already demonstrated that the dynamics in presence of higher-order interactions can be significantly different from that of systems where interactions are exclusively pairwise[2,8–11]. How to infer and model higher-order interactions is then crucial for understanding the dynamics and functioning of complex systems[12,13]. While in complex networks, the reconstruction problem, also known as the inverse problem, i.e., determining the network from the dynamics of a system, has been dealt with different techniques[14], the question on how to infer connectivity in the presence of higher-order interactions is still open.

Concerning reconstruction in complex networks, two different types of approaches, which target either the functional or the structural connectivity of the system, have been developed[14]. Functional connectivity accounts for the statistical dependencies that develop among the network units as the result of their dynamical interactions. As such, it may depend on the function and state of the system, and the same physical system may display different functional connectivities depending on the dynamical state where it is operating. On the contrary, the structural connectivity represents how the units of a system physically interact with each other. Functional networks are typically constructed from the network temporal evolution by evaluating statistical similarity measures[15], Granger causality[16,17], or transfer entropy among the signals of the different network units[18,19], or using Bayesian inference methods[20]. The underlying structural connectivity of a network can be, instead, obtained from the network response to external perturbations[21], from its synchronization with a copy containing adaptive links[22,23], or from the solution of optimization problems based on measurements of node time series, when the functional form of the node dynamics is known[24–26].

The reconstruction problem in the presence of higher-order interactions is more convoluted. Recently, the fundamental distinction between *higher-order mechanisms*, i.e., the presence of higher-order terms in the microscopic structure of the interactions, and *higher-order behaviors*, i.e., the emergence of higher-order correlations in the dynamical behavior of a system, has been pointed out[27]. The relationship between these two is not trivial as higher-order behaviors do not necessarily rely on higher-order mechanisms. As an example, three-node correlations can appear even in systems with pairwise interactions only, due to the particular structure of the network, rather than to the presence of genuine three-body interactions. However, for the identification of higher-order interactions, techniques that go beyond pairwise statistics are required. For instance, information-theoretic approaches to study multivariate time series (of node activities) based on hypergraphs[28], higher-order predictability measures

[1]Dipartimento di Fisica ed Astronomia, Università di Catania, Catania, Italy. [2]Network Science Institute, Northeastern University London, London E1W 1LP, UK. [3]Department of Electrical Electronic and Computer Science Engineering, University of Catania, Catania, Italy. [4]Department of Mathematics and Computer Science, University of Catania, Catania, Italy. [5]School of Mathematical Sciences, Queen Mary University of London, London E1 4NS, UK. [6]INFN, Catania, Italy. [7]Complexity Science Hub, Josefstädter Strasse 39, A 1080 Vienna, Austria. ✉e-mail: mattia.frasca@dieei.unict.it

(such as generalizations of Granger causality and partial information decomposition)[29], or simplicial filtration procedures[30] have been proposed to extract important information on higher-order behaviors that otherwise would not be visible to standard, i.e., network-based, analysis tools. Higher-order behaviors, which are likely due to the presence of higher-order mechanisms, can be identified by recently introduced techniques to assess the statistical significance of certain hyperlinks[31,32]. Other statistical approaches to the problem are based on Bayesian methods and have been used to construct hypergraphs directly from pairwise measurements (link activities), even in cases where the higher-order interactions are not explicitly encoded[33,34]. Statistical inference and expectation maximization are also at the basis of a method recently developed to reconstruct higher-order mechanisms of interaction in simplicial SIS spreading and Ising Hamiltonians with two- and three-spin interactions[35]. However, this method can only be applied to binary time series data produced by discrete two-state dynamical models.

In this paper, we propose an optimization-based approach to infer the high-order structural connectivity of a complex system from its time evolution, which works in the case of the most general continuous-state dynamics, i.e., when node variables are not restricted to take binary values. Namely, we consider a system of ordinary differential equations (ODE) describing a set of dynamical units (nodes) coupled through pairwise and higher-order interactions. We assume that the local dynamics and the functional form of the interactions are known[36] or identifiable[37,38], and we propose a method to extract the topology of such interactions by solving an optimization problem based on the measurement of the time evolution of the node variables. The method relies on the derivation of a set of algebraic equations for the parameters that appear in the ODE model under consideration. Similar approaches have already been adopted to solve identification problems in other contexts[25,26,39–41]. The unknown terms in the algebraic equations are determined through least square minimization, possibly applying regularization methods. Such unknown terms can either describe the ways in which the variables are combined together (namely, the coefficients of the different linear and nonlinear terms effectively appearing in the equations of a single dynamical unit[39]), or the way in which variables from different units are effectively coupled (namely, the weights of the pairwise interactions in a network of coupled dynamical units[25,26,40,41]). In the first case, the approach aims at deriving the governing equations of an unknown dynamical system from measurements of its state[39], whereas in the second case, the functional form of the local dynamics is supposed to be known and the approach aims at reconstructing the interactions among the units[25,26,40]. In both cases, the problem is sparse, as various dynamical systems can be modeled by using a small number of nonlinear terms from the large dictionary of possible nonlinear functions, and the interactions among the units of a complex system are often mapped into a sparse network. This enables the use of optimization problems based on compressive sensing, such as the Lasso method, to solve this type of problem. While previous works[25,26,40] have addressed the reconstruction problem of pairwise networks, the method we propose here deals with the more general case of structures with interactions of any order. With two concrete applications, we will show that our method can effectively reconstruct which nodes are interacting in pairs and which in groups of three or more nodes.

## Results

### Reconstructing pairwise and higher-order interactions

As a general model of a dynamical system of $N$ nodes coupled through pairwise and higher-order interactions, we consider the following set of equations:

$$\dot{\mathbf{x}}_i = \mathbf{f}_i(\mathbf{x}_i) + \sum_{d=1}^{D}\sum_{j_1,\ldots,j_d=1}^{N} a_{ij_1\cdots j_d}^{(d)}\mathbf{g}^{(d)}(\mathbf{x}_i,\mathbf{x}_{j_1},\ldots,\mathbf{x}_{j_d}), \quad (1)$$

with $i = 1, \ldots, N$. Here $\mathbf{x}_i(t) \in \mathbb{R}^n$ is the state vector of unit $i$, $\mathbf{f}_i : \mathbb{R}^n \to \mathbb{R}^n$ is the nonlinear function describing the local dynamics at node $i$, while $\mathbf{g}^{(d)} : \mathbb{R}^{n \times (d+1)} \to \mathbb{R}^n$ are the nonlinear functions of order $d$, modeling interactions in groups of $d+1$ nodes, with $d = 1, \ldots, D$. The topology of the $(d+1)$-body interactions is encoded in the tensor $\mathbf{A}^{(d)}$, whose components are real numbers denoting the strengths of the interactions. The components with two or more equal indices vanish because the interaction is in this case described by a lower rank tensor (or there is no interaction if all indices are equal). Therefore, for a given $i$, the number of non-zero entries is at most $N-1$ for $\{a_{ij}^{(1)}\}$, $(N-1)(N-2)$ for $\{a_{ijk}^{(2)}\}$, and so on.

Here we want to infer the complete structural connectivity of a dynamical system, which means we want to reconstruct, not only the entries of the adjacency matrix $\mathbf{A}^{(1)} = \{a_{ij}^{(1)}\}$ from the knowledge of the evolution of the state variables $\mathbf{x}_1(t), \ldots, \mathbf{x}_N(t)$, but also the higher-order interactions encoded by the tensors $\mathbf{A}^{(2)} = \{a_{ijk}^{(2)}\}, \ldots, \mathbf{A}^{(D)} = \{a_{ij_1\cdots j_d}^{(d)}\}$. In doing this, we do not assume any specific structure for the tensors, except that their maximum rank is $D+1$. Conversely, we assume that the functions $\mathbf{f}_i$ and $\mathbf{g}^{(1)}, \mathbf{g}^{(2)} \ldots, \mathbf{g}^{(D)}$, are known. This is reasonable as the local dynamics of many real-world complex systems, as well as the functional forms of their interactions, have been well identified. For instance, well-established mathematical models that describe the dynamics of neurons and synapses, or the growth of a biological species when in isolation, or when in interactions with other species, are available. In the absence of such models, we assume instead that, prior to the structural connectivity reconstruction, the model of the isolated dynamics of a single unit, a pair, a group of three units, etc. can be derived using proper identification techniques[37].

Our reconstruction technique works as follows. Suppose we have access to a sample of $M+1$ measurements of the variables $\mathbf{x}_1(t), \ldots, \mathbf{x}_N(t)$ at times $t_m$ equal to $m\Delta t$, with $\Delta t$ a (constant) sampling interval and $m = 0, \ldots, M$. The idea here is to find the entries of the tensors $\mathbf{A}^{(d)}$, $d = 1, \ldots, D$, which minimize the discrepancy between the sampled values and the trajectories generated by the model. We will discuss our method, first under the assumption that it is possible to directly measure the derivatives $\dot{\mathbf{x}}_i(t)$ at times $t_m$, and, later on, under the less restrictive assumption that we have access only to the variables $\mathbf{x}_1(t), \ldots, \mathbf{x}_N(t)$. In the latter case, we have to approximate the derivatives $\dot{\mathbf{x}}_i$ in Eq. (1) making use of the sampled values. In both cases, from system (1) at time $t_m$ the following equations hold

$$\begin{aligned}
\mathbf{z}_i^m &= \mathbf{f}_i(\mathbf{x}_i^m) \\
&+ \sum_{d=1}^{D}\sum_{j_1,\ldots,j_d=1}^{N} a_{ij_1\cdots j_d}^{(d)}\mathbf{g}^{(d)}(\mathbf{x}_i^m,\mathbf{x}_{j_1}^m,\ldots,\mathbf{x}_{j_d}^m),
\end{aligned} \quad (2)$$

where $i = 1, \ldots, N$, $m = m_{\min}, \ldots, m_{\max}$, $\mathbf{x}_i^m$ is a short notation for $\mathbf{x}_i(t_m)$, and $\mathbf{z}_i^m$ is the measured value of the derivative $\dot{\mathbf{x}}_i$ at time $t_m$, in the first case, or its approximation by a suitable finite difference method, in the second case. Here, $m_{\min} \geq 0$ and $m_{\max} \leq M$ are integers that depend on the approximation adopted (see Methods).

Now, let

$$\mathbf{y}_i^m := \mathbf{z}_i^m - \mathbf{f}_i(\mathbf{x_i}^m)$$

and $\mathbf{Y}_i = [\mathbf{y}_i(m_{\min})^\top, \ldots, \mathbf{y}_i(m_{\max})^\top]^\top$. Let us, then, define the vector $\mathcal{A}_i$ containing the non-trivial elements of the tensors $\mathbf{A}^{(d)}$, $d = 1, \ldots, D$, namely the quantities we want to reconstruct for each node $i$

$$\begin{aligned}
\mathcal{A}_i &\equiv [(\mathcal{A}_i^{(1)})^\top, (\mathcal{A}_i^{(2)})^\top, \ldots, (\mathcal{A}_i^{(D)})^\top]^\top = \\
&= [a_{i1}^{(1)}, \ldots, a_{iN}^{(1)}, a_{i12}^{(2)}, \ldots, a_{i,N-1,N}^{(2)}, \ldots, a_{i,1,\ldots,D}^{(D)}, \ldots, a_{i,N-D+1,\ldots,N}^{(D)}]^\top.
\end{aligned}$$

Then, from Eq. (2), we get:

$$\mathbf{Y}_i = \Phi_i \mathcal{A}_i \tag{3}$$

for $i = 1, ..., N$, with

$$\Phi_i = \begin{bmatrix} \mathbf{g}_{i,1}^{(1)}(m_{\min}) & \cdots & \mathbf{g}_{i,i-1}^{(1)}(m_{\min}) & \mathbf{g}_{i,i+1}^{(1)}(m_{\min}) & \cdots & \mathbf{g}_{i,N}^{(1)}(m_{\min}) & \mathbf{g}_{i,1,2}^{(2)}(m_{\min}) & \cdots & \mathbf{g}_{i,N-D+1,...,N}^{(D)}(m_{\min}) \\ \vdots & & & & & & & & \vdots \\ \mathbf{g}_{i,1}^{(1)}(m_{\max}) & \cdots & \mathbf{g}_{i,i-1}^{(1)}(m_{\max}) & \mathbf{g}_{i,i+1}^{(1)}(m_{\max}) & \cdots & \mathbf{g}_{i,N}^{(1)}(m_{\max}) & \mathbf{g}_{i,1,2}^{(2)}(m_{\max}) & \cdots & \mathbf{g}_{i,N-D+1,...,N}^{(D)}(m_{\max}) \end{bmatrix} \tag{4}$$

where we introduced the following short notation: $\mathbf{g}_{i,j_1,...,j_d}^{(d)}(m) := \mathbf{g}^{(d)}(\mathbf{x}_i^m, \mathbf{x}_{j_1}^m, ..., \mathbf{x}_{j_d}^m)$.

For each node $i$ we need to identify $H = N - 1 + (N - 1)(N - 2) + ... + (N - 1) \cdots (N - D)$ terms, corresponding to the entries of $\mathcal{A}_i$. Let $\tilde{M} = (m_{\max} - m_{\min} + 1) n_g$, where $n_g$ represents the number of non-trivial components of the coupling functions, then $\Phi_i \in \mathbb{R}^{M \times H}$.

When $\tilde{M} < H$, the system of Eq. (3) is underdetermined and multiple solutions may exist[14]. Conversely, when $\tilde{M} \geq H$ and the matrices $\Phi_i$ are full rank, i.e., their columns are linearly independent, the system in Eq. (3) is not underdetermined and can be solved in the least square sense[14]. In this case, the tensors $\mathcal{A}_i$ are uniquely determined.

Solving system (3) for the unknowns $\mathcal{A}_i$, therefore, allows one to reconstruct all interactions of node $i$, such that the whole structural connectivity can be inferred by repeating the calculations for all nodes, $i = 1, ..., N$. Notice that Eq. (3) maps the problem of the reconstruction of the higher-order interactions into that of solving a system of algebraic equations in the unknown variables given by the $H$ entries of $\mathcal{A}_i$.

In general, the identifiability[38,42,43] of the problem mainly depends on the collected data, and is difficult, if not impossible, to establish a priori conditions for it. We, therefore, formulate the identification task in terms of different optimization problems, depending on the type of structure (weighted/unweighted, directed/undirected) and also on the features of the available data. Namely, we propose to use, together with the ordinary least squares (OLS), two other different optimization methods, the Signal Lasso (SL)[26] and the non-negative least squares (NNLS)[44]. The latter two methods are based on a regularization of the problem and provide an optimal solution even when $\tilde{M} < H$. In general, all the methods will show an improvement in their results as the number $M$ of observations increases, provided that the added measurements are able to sample different states of the system dynamics, thus bringing new information.

We will now show that our approach is able to successfully reconstruct the full set of interactions at any order for completely different dynamics. For these reasons, we will focus on the dynamics of microbial ecosystems and on that of coupled chaotic oscillators. The two considered case studies will also demonstrate that our method works for the reconstruction of both hypergraphs and simplicial complexes, no matter whether the underlying structure is undirected or directed, unweighted or weighted.

## Lotka–Volterra dynamics on weighted directed hypergraphs

In our first application, we focus on the dynamics of microbial ecosystems. These consist of species that may engage in diverse relationships, either cooperative, such as the transfer of complementary metabolites, or antagonistic, such as competition for a resource[45]. The validation of community-wide interactions in microbial communities is a far from trivial problem, faced both with experimental approaches[46] and through the use of mathematical modeling[47]. The problem is further complicated by potential higher-order interactions, which play a role in stabilizing diverse ecological communities and maintaining species coexistence[1,48,49]. Here, we model a microbial

ecosystem of $N$ species as a hypergraph of $N$ coupled Lotka–Volterra type equations[50] including both pairwise and three-body interactions:

$$\dot{x}_i = f_i(x_i) + \sum_{j=1}^{N} a_{ij}^{(1)} x_i x_j + \sum_{j=1}^{N} \sum_{k=j+1}^{N} a_{ijk}^{(2)} x_i x_j x_k \tag{5}$$

with $i = 1, ..., N$.

The variable $x_i$ represents the abundance of species $i$. The local dynamics of $x_i$ is governed by the logistic function $f_i(x_i) = r_i x_i (1 - \frac{1}{k_i} x_i)$ where $r_i$ and $k_i$ are the growth rate and the carrying capacity. The pairwise interactions between species are encoded in the real coefficients of the $N \times N$ weighted matrix $A^{(1)} = \{a_{ij}^{(1)}\}$ with at most $N(N-1)$ non-zero elements, while the three-body interactions in the real coefficients of the $N \times N \times N$ weighted tensor $A^{(2)} = \{a_{ijk}^{(2)}\}$ with at most $N \binom{N}{2}$ non-zero elements. This is because when the interaction does not depend on the permutation of the indices, as is the case of Lotka–Volterra type models, the number of non-zero entries of tensor $A_i^{(d)}$ is $\binom{N}{d}$. Equation (5) is in the form of Eq. (1) with $g^{(1)}(x_i, x_j) = x_i x_j$ and $g^{(2)}(x_i, x_j, x_k) = x_i x_j x_k$. As an example, we consider the system of $N = 7$ species with four cooperative ($a_{ij}^{(1)} > 0$) and four antagonistic ($a_{ij}^{(1)} < 0$) pairwise interactions, studied in ref. 47 and shown in Fig. 1a with blue and red arrows, respectively. In addition to these pairwise interactions, we have included two cooperative three-species interactions, shown as double arrows in the hypergraph in Fig. 1a. These respectively correspond to a contribution to the dynamics of $x_2$ given by $a_{237}^{(2)} x_2 x_3 x_7$ and one to $x_4$ given by $a_{416}^{(2)} x_4 x_1 x_6$, with $a_{237}^{(2)} = 0.0062$ and $a_{416}^{(2)} = 0.0016$[1]. The other system parameters, i.e., the values of $r_i, k_i, i = 1, ..., 7$, and the initial conditions have been chosen as in ref. 47. Namely, growth rates $r_i$ for all species have been randomly selected from a uniform distribution in the interval (0, 1), similarly, the carrying capacities $k_i$ are sampled from a uniform distribution in the interval (1, 100), and the initial conditions $x_i(0)$ are integers sampled in the interval (10, 100).

Under these conditions, as shown by the time evolution of the variables $x_i(t)$, with $i = 1, ..., 7$, reported in Fig. 1b, the microbial ecosystem typically converges to a stable equilibrium point corresponding to the coexistence of six species over seven. To feed our reconstruction algorithm, we focused on the time window $[0, t_{\max}]$ with $t_{\max} = 20$, and we sampled the seven trajectories at $M$ regular intervals of size $\Delta t = t_{\max}/M$. We then used the samples to calculate $\mathbf{Y}_i$ and $\Phi_i$ from Eq. (4). At this point, we adopted OLS method based on the complete orthogonal decomposition (see "Methods" and ref. 51), which provides the optimal solution of minimal discrepancy $E$ between the measurements $\mathbf{Y}_i$, and the corresponding values produced by the system $\Phi_i \mathcal{A}_i$, where

$$E = \sum_{i=1}^{N} E_i, \text{with } E_i \equiv \| \mathbf{Y}_i - \Phi_i \mathcal{A}_i \|_2^2. \tag{6}$$

The total number of parameters is $NH$, while the total number of sampled values is $nNM$ (in this case $n = 1$). The minimum of the sum is

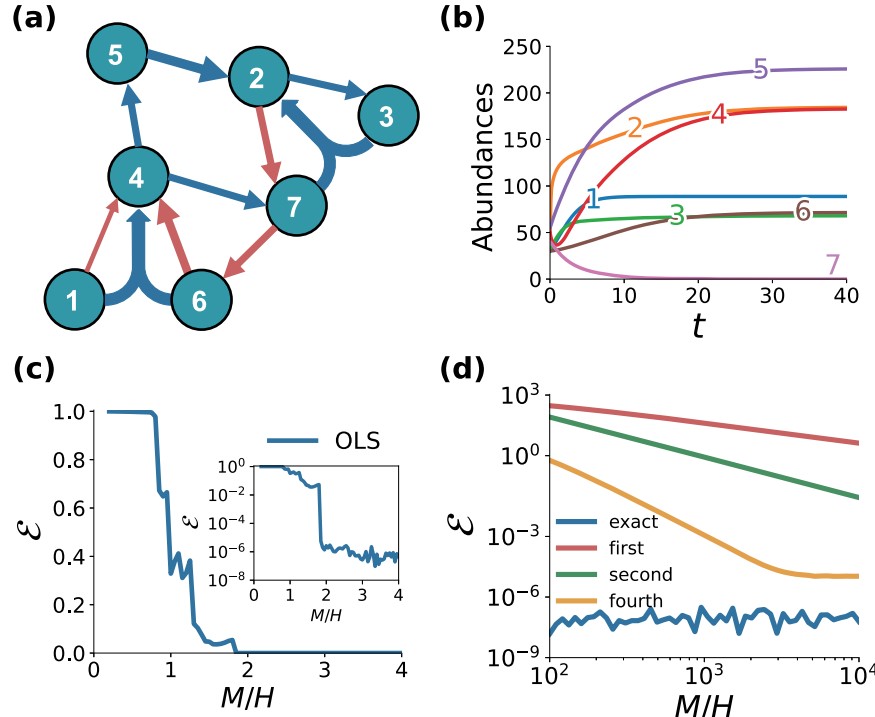

**Fig. 1 | Reconstructing higher-order interactions in a microbial ecosystem.**
**a** The underlying weighted hypergraph of a Lotka–Volterra system with $N = 7$ species and two- and three-body interactions, which we want to reconstruct from (**b**) the time evolution of the seven species abundance $x_i(t)$. **c** Quality of the reconstruction is measured by reporting the error $\mathcal{E}$ as a function of the ratio between the length $M$ of the trajectories and the number $H$ of interactions to reconstruct. **d** Error $\mathcal{E}$ for the various approximations of the derivatives.

obtained by minimizing each term $E_i$ separately. The complete orthogonal decomposition provides the solution of minimal $\ell^2$ norm when there is more than one minimizer.

To quantify the accuracy of the reconstruction of the interactions at any order, we compare the estimation $\widehat{\mathcal{A}}_i$ with the true values of the couplings, $\mathcal{A}_i$, for each $i$, evaluating the reconstruction error $\mathcal{E}$ as:

$$\mathcal{E}^2 = \frac{\sum_{i=1}^{N} \left\| \mathcal{A}_i - \widehat{\mathcal{A}}_i \right\|_2^2}{\sum_{i=1}^{N} \left\| \mathcal{A}_i \right\|_2^2} \tag{7}$$

Figure 1c shows $\mathcal{E}$ as a function of $M/H$, under the assumption that the derivatives are available. Different values of $M/H$ have been obtained by changing the number of measurements $M$, while the number of unknown coefficients we want to determine is $H = N - 1 + (N-1)(N-2)/2 = 21$, where the factor 2 in the denominator is due to the symmetry of the interaction terms in Eq. (5), and contributes to reducing the number of unknowns and consequently the computational cost. The results indicate that our approach correctly reconstructs both pairwise and three-body interactions of the hypergraph, as the error drops down when $M/H \approx 1.8$.

If only the values of the trajectories are available at sampling times, then the derivatives have to be estimated numerically, for example, by finite difference approximation. We used three different formulas for the computation of the approximate derivatives, respectively of order of accuracy 1, 2, and 4 (see Methods). As a consequence of this approximation, the identification improves when increasing the sampling size, mainly because of the improvement in the accuracy of the numerical derivatives as the sampling time interval decreases. The results are reported in Fig. 1d. As expected, the fourth-order method provides a smaller error $\mathcal{E}$ for a given sampling data size $M$. The level of approximation is limited by the accuracy in the measurement of the trajectories. In particular, because of numerical

cancellation, the relative error in the evaluation of the derivatives is always larger than the relative uncertainty in the trajectories.

**Coupled Rössler oscillators on undirected simplicial complexes**
As a second case study, we analyze the following system of Rössler oscillators coupled with pairwise and three-body interactions:

$$\dot{x}_i = -y_i - z_i + \sigma_1 \sum_{j=1}^{N} a_{ij}^{(1)} g^{(1)}(x_i, x_j)$$
$$+ \sigma_2 \sum_{j=1}^{N} \sum_{k=j+1}^{N} a_{ijk}^{(2)} g^{(2)}(x_i, x_j, x_k), \tag{8}$$
$$\dot{y}_i = x_i + a y_i,$$
$$\dot{z}_i = b + z_i(x_i - c),$$

where $g^{(1)}(x_i, x_j) = x_j - x_i$ and $g^{(2)}(x_i, x_j, x_k) = x_j^2 x_k + x_j x_k^2 - 2x_i^3$. As for the underlying topology of the interactions, namely the components of tensors $A^{(1)}$ and $A^{(2)}$, we consider simplicial complexes constructed as follows. We start from the so-called Zachary karate club, which is a system originally described in terms of an undirected graph with $N = 34$ nodes and 78 links[52]. Since the links form 45 triangles, we can represent the system as a simplicial complex by turning a randomly chosen fraction $\delta$ of the triangles into two-dimensional simplices[4]. By considering different values of $\delta$, we can then tune the percentage of the nodes forming a triangle which are effectively involved in a three-body interaction rather than in three, separate, pairwise interactions only. In Fig. 2, we report the results obtained for $\delta = 1$. Similar results are obtained for other values of $\delta$. The simplicial complex considered is shown in Fig. 2a. Notice that, in this case, the components of tensors $A^{(1)}$ and $A^{(2)}$ can either be 0 or 1, as they describe an unweighted structure. The strength of the interactions in Eq. (8) is parameterized by the constants $\sigma_1 > 0$, $\sigma_2 > 0$, which are supposed to be known. Here, we choose $\sigma_1 = 10^{-4}$, and $\sigma_2 = 10^{-5}$.

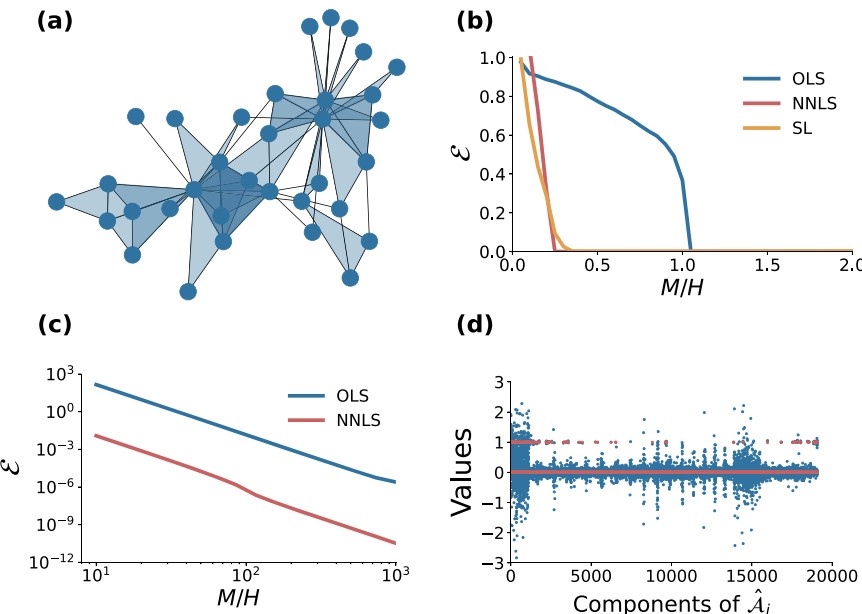

**Fig. 2 | Testing the reconstruction method on a system of $N = 34$ coupled Rössler oscillators. a** The underlying simplicial complex consists of 78 links and 45 2-simplices. **b** Reconstruction error $\mathcal{E}$ defined in Eq. (7) as a function of $M/H$ when derivatives are known. **c** The computed values of the components of the arrays $\widehat{\mathcal{A}}_i, i = 1, \ldots, N$ for $M/H = 34$ for the OLS method (blue dots) and for the NNLS method (red dots). **d** Reconstruction error $\mathcal{E}$ as a function of $M/H$ when derivatives are not known and a fourth-order approximation of the derivatives is used.

In spite of the fact that the original structure is undirected, in our reconstruction we treat the problem as if the structure was directed. This means that for a given pair of connected nodes $i$ and $j$ we consider both terms $a_{ij}^{(1)}$ and $a_{ji}^{(1)}$ in the adjacency matrix as independent unknowns to be determined. Analogously, taking into account that in Eq. (8) the only non-zero coefficients of $A^{(2)}$ are those with the third index larger than the second one, for a 3-node simplex $(i, j, k)$ with $i < j < k$ we have three independent unknowns $a_{ijk}^{(2)}, a_{jik}^{(2)}$, and $a_{kij}^{(2)}$.

In this way, the discrepancies $E_i$ appearing in Eq. (6) can be minimized independently, therefore simplifying the reconstruction problem. When $M$ is large enough, so that the error is almost zero, the reconstructed tensors are indeed almost symmetric, i.e., $a_{ij}^{(1)} \approx a_{ji}^{(1)}$ for the first order and $a_{ijk}^{(2)} \approx a_{jik}^{(2)} \approx a_{kij}^{(2)}$ for the second order.

Considering that the interactions to be reconstructed are, in this case, unweighted, the number of the different higher-order networks we can obtain is finite and equal to $2^{NH}$, where $NH$ is the number of all possible interactions among the $N$ nodes. An exhaustive search over all possible structures is prohibitive for even moderate values of $N$. For such a reason we shall resort to methods for continuous weights, possibly making use of the a priori information of the binary nature of the tensors in order to obtain a more effective reconstruction.

We have adopted three different methods to solve Eq. (3) when the derivative is exactly known. In the first method, namely OLS, we minimize the least square norm of the difference between $\mathbf{Y}_i$ and $\Phi_i \mathcal{A}_i$, similarly to what we did in the microbial ecosystem example. The results, indicated by the blue line in Fig. 2b, show that the method correctly reconstructs the simplicial complex when $M/H$ approaches one. In the second method, namely NNLS, we perform the minimization of the least square error under the additional constraint that the elements of $\mathcal{A}_i$ are non-negative: $\min_{\mathcal{A}_i \geq 0} \| \mathbf{Y}_i - \Phi_i \mathcal{A}_i \|_2$. The red curve in Fig. 2b indicates that, including such a priori information on the nature of the interactions in the optimization problem, reduces the values of $M/H$ necessary for the reconstruction. Lastly, we extend SL[26] to deal with higher-order interactions. Namely, we consider the following optimization: $\min(\| \mathbf{Y}_i - \Phi_i \mathcal{A}_i \|_2^2 / 2 + \alpha \| \mathcal{A}_i \|_1 + \beta \| \mathcal{A}_i - \mathbf{1}_H \|_1)$, where the penalty function includes, together with the square of the 2-norm of the difference between $\mathbf{Y}_i$ and $\Phi_i \mathcal{A}_i$, a regularization term that induces sparsity of the solution by penalizing the non-zero elements of $\mathcal{A}_i$, and another term to shrink the estimates to one, by penalizing the elements different from one (case of unweighted graph). The effect of the two penalty terms can be easily understood graphically by plotting $\mathrm{argmin}\{(\hat{x} - x)^2 / 2 + \alpha|x| + \beta|x - 1|\}$ as function of $\hat{x}$ (see Fig. 1 in ref. 26). As indicated by the orange curve in Fig. 2b, this method provides successful reconstruction with a performance similar to the NNLS (red line). In conclusion, the last two methods are able to fully reconstruct the structure of the simplicial complex, with a smaller sample size than that of OLS.

When the derivatives are not available, we need to approximate them. Here, we use a fourth-order approximation and consider as a first example $M/H = 34$. In Fig. 2c, we show the values of the components of the arrays $\widehat{\mathcal{A}}_i, i = 1, \ldots, N$ obtained with the OLS (blue dots) and the NNLS (red dots) methods. For the OLS method, the value of $M/H$ is still too low and values of the components of $\widehat{\mathcal{A}}_i$ spanning from $-3$ to $3$ are obtained; on the contrary, for the NNLS method the components of $\widehat{\mathcal{A}}_i$ are concentrated around the two possible values (0 and 1), correctly estimating the true values of the coefficients appearing in the tensors $A^{(1)}$ and $A^{(2)}$. Now we systematically analyze the dependence of the error on the ratio $M/H$. In Fig. 2d, we report the results for the two more computationally efficient methods, namely OLS (blue curve) and NNLS (red curve). Although larger values of $M/H$ are required when the derivatives are not available, we find that, due to the approximation used, the error $\mathcal{E}$ decreases as a power-law of $M/H$ with an exponent $-4$ in both cases, with NNLS consistently giving better results than OLS.

## Discussion

The collective behavior of a complex system is the result of the interactions taking place among its units. Recent evidence suggests that these interactions occur not only between pairs of agents but also within groups involving multiple units. Consequently, fully reconstructing the high-order structural connectivity of a complex system becomes crucial. In this paper, we have presented an optimization-based framework aimed at tackle this problem. Our approach takes as input the measurements of the states of the nodes at different times, yielding a linear system for the unknowns of the problem, namely the

**Table 1 | Parameters and initial conditions used to generate the data of the example of the microbial ecosystem**

| Node | 1 | 2 | 3 | 4 | 5 | 6 | 7 |
|---|---|---|---|---|---|---|---|
| $R_i$ | 0.6099 | 0.6177 | 0.8594 | 0.8055 | 0.5767 | 0.1829 | 0.2399 |
| $K_i$ | 88.7647 | 3.8387 | 49.5002 | 17.6248 | 97.8894 | 71.5568 | 50.5467 |
| $x_i(0)$ | 30 | 45 | 32 | 50 | 55 | 30 | 40 |

entries of the adjacency tensors encoding the pairwise and higher-order interactions. Despite the complexity increases with respect to both system size and order of interactions, the linear nature of our approach allows for the application of various numerical methods specifically designed for such problems[53].

A crucial aspect of our approach relies on understanding the dynamics of individual units and the coupling functions at different orders. Prior to reconstructing the interactions, it is necessary to derive the functional form of the unit dynamics and coupling functions. This can be achieved through either first principles or targeted experiments. Importantly, these experiments can be conducted on small-scale systems using an iterative procedure. The process begins with identifying the dynamical equations of an isolated unit and gradually incorporating higher orders of interaction. At each iteration, when aiming to determine the functional form of the $d$-th order interaction, namely $g^{(d)}$, an ensemble of $d+1$ interacting units is considered. The temporal evolution of the variables in this system is measured, and identification techniques[37] are employed to obtain the governing equations. In the previous steps, $f$, $g^{(1)}$, ..., $g^{(d-1)}$ have been determined, leaving $g^{(d)}$ as the remaining unknown. The process of accomplishing these steps for a real system remains open, likely requiring tailored solutions specific to the nature of the system under investigation.

In this paper, we have demonstrated the effectiveness of our reconstruction method through a series of practical applications. We have shown that with our approach we can detect which nodes are interacting in pairs and which in groups of three or more nodes, enabling the reconstruction of hypergraphs and simplicial complexes. Importantly, the technique can be applied to both undirected and directed structures, as well as unweighted and weighted networks. Directed topologies have the advantage that the discrepancy function $E$ defined in Eq. (6) can be minimized node by node, i.e., by considering each term $E_i$ separately. Undirected topologies with the same number of nodes contain fewer unknowns, but the discrepancy function $E$ cannot be minimized node by node because of the symmetry constraints. For instance, the value of $a_{ij}^{(1)}$ appearing in $E_i$ needs to be equal to $a_{ji}^{(1)}$ in $E_j$. When $M$ is large enough, and the signal is not affected by noise, application of the directed framework is able to recover the correct network, and thus the symmetry of the coefficients, as illustrated by the case study of coupled Rössler oscillators. In other situations, for example, in the presence of noise or with a more sparse data sampling, strict symmetry might be violated by a straightforward application of the directed framework. In such cases, the symmetry of the coefficients can be imposed by using other techniques, e.g., by introducing a penalization term on the lack of symmetry.

In our work, we have considered relatively small systems, in which there is no a priori knowledge of their structure. For large systems it is not feasible to consider all possible interactions, since the number of unknowns grows as $N^D$, where $D$ is the maximum order of interaction we consider, making the problem practically intractable. In the case some basic information is available, for example, if one knows the graph describing the pairwise backbone of the system, then one could limit, for instance, the search of higher-order interactions of dimension 2 among the triangles of the graph.

Finally, we note that, since the reconstruction method proposed in this paper is based on the numerical approximation of the derivatives from the data, in the presence of noisy signals the method cannot be directly applied as is. In such a case one can resort to the more traditional approach based on the minimization of the discrepancy between the noisy measured trajectories and the computed ones that depend on the hypergraph parameters. However, this technique leads to a strongly nonlinear least square problem, which may be computationally very expensive. An alternative approach consists in the application of our method after a suitable pre-processing of the measured data. The latter approach is currently under investigation.

In conclusion, we have shown that our technique can be useful to understand and predict the behavior of microbial ecosystems and coupled nonlinear oscillators. We hope that it can shed new light on a variety of physical phenomena where higher-order interactions have a fundamental role.

## Methods

### Representing higher-order interactions

Higher-order interactions are mathematically represented either by hypergraphs or by simplicial complexes[4]. Given a set $\mathcal{N}$ of $N$ nodes, an undirected (directed) hypergraph $\mathcal{H}$ is an aggregate of $N_h$ hyperedges, $\mathcal{H} = \{e_1, e_2, \ldots, e_{N_h}\}$, where each hyperedge $e_i$, with $i = 1, \cdots, N_h$, is an unordered (ordered) subset of two or more nodes, respectively describing pairwise and group interactions. A simplicial complex $\mathcal{S}$ is also an aggregate of objects, called simplices, that generalize edges and can be of different dimensions. A $d$-simplex $\sigma$, or simplex of dimension $d$ is, in its simplest definition, a collection of $d+1$ nodes. In this way, a 0-simplex is a node, a 1-simplex is an edge, a 2-simplex $(i, j, k)$ is a two-dimensional object made by three nodes, usually called a (full) triangle, a 3-simplex is a tetrahedron, i.e., a three-dimensional object and so on. A simplicial complex $\mathcal{S}$ on the set of nodes $\mathcal{N}$, is a collection of $N_s$ simplices, $\mathcal{S} = \{\sigma_1, \sigma_2, \ldots, \sigma_{N_s}\}$, with the extra requirement that, for any simplex $\sigma \in \mathcal{S}$, all the simplices $\sigma'$ with $\sigma' \subset \sigma$, i.e., all the simplices built from subsets of $\sigma$, are also contained in $\mathcal{S}$. Due to this constraint, simplicial complexes are a very particular type of hypergraphs[7]. The connectivity of a hypergraph or of a simplicial complex of $N$ nodes is fully described by the set of tensors $A^{(1)}, A^{(2)}, \cdots, A^{(D)}$ whose components are real numbers denoting the strengths of the interactions at different orders. In this article we focus on higher-order structures with $D = 2$, i.e., on systems with pairwise and three-body interactions described by the matrix $A^{(1)} = \{a_{ij}^{(1)}\}$ and the tensor $A^{(2)} = \{a_{ijk}^{(2)}\}$.

### Integration of equations and approximation of derivatives

In order to create the two case studies, we choose a set of two- and three-body interactions (i.e., the tensors $A^{(1)}$ and $A^{(2)}$) that we aim to identify, and compute the trajectories by accurate numerical integration of system (1). The integration of the system of ODE's has been performed by the MATLAB routine $\circ$de45 with absolute and relative tolerance equal to $10^{-12}$. In the simulation of the microbial ecosystem, we adopted the values of the parameters reported in Table 1, while in the simulation of coupled Rössler oscillators we set $a = 0.2$, $b = 0.2$, $c = 9$ and we used initial conditions randomly sampled from uniform distributions with the following ranges: $x_i(0) \in [-3, 3]$, $y_i(0) \in [-3, 3]$, $z_i(0) \in [-1, 7]$.

The derivative of each component has been computed by standard finite difference discretization. In particular, the approximation of the derivative of the generic function $x(t)$, computed at time $t^m = m\Delta t$, is given by the following expressions

$$\dot{x} = \frac{x^m - x^{m-1}}{\Delta t} \qquad\qquad + O(\Delta t)$$
$$\dot{x} = \frac{x^{m+1} - x^{m-1}}{2\Delta t} \qquad\qquad + O(\Delta t^2)$$
$$\dot{x} = \frac{-x^{m+2} + 8x^{m+1} - 8x^{m-1} + x^{m-2}}{12\Delta t} \qquad + O(\Delta t^4)$$

In the previous expressions, we have $m = 0, \ldots, M-1$, i.e., $m_{\min} = 0$ and $m_{\max} = M-1$ for the approximation of order one; $m = 1, \ldots, M-1$, i.e.,

$m_{min} = 1$ and $m_{max} = M - 1$ for the order two; and $m = 2, ..., M - 2$, i.e., $m_{min} = 2$ and $m_{max} = M - 2$ for the order four. When the derivatives are supposed to be known, then $m = 0, ..., M$, i.e., $m_{min} = 0$ and $m_{max} = M$.

## Optimization methods

OLS method has been implemented using the MATLAB function `lsqminnorm`, which is based on the complete orthogonal decomposition. `lsqminnorm(A, b)`, where $A \in \mathbb{R}^{n \times m}$, solves the minimization problem

$$\min_{\mathbf{x}} \| A\mathbf{x} - \mathbf{b} \|_2.$$

If the vector of minimal norm is unique, i.e., when the rank of A is greater or equal $m$, then the method is essentially equivalent to standard linear least square. If there are more solutions for vector $x \in \mathbb{R}^m$, the function returns the one with the minimal $\ell^2$ norm.

For NNLS we used the MATLAB function `lsqnonneg`, which solves $\ell^2$ minimization problem with the constraint that the solution should be non negative. `lsqnonneg(A, b)` solves the problem

$$\min_{\mathbf{x}} \| A\mathbf{x} - b \|_2$$

with the constraints $x \ge 0$.

For SL we wrote a custom MATLAB function based on the algorithm described in ref. 26.

### Reporting summary

Further information on research design is available in the Nature Portfolio Reporting Summary linked to this article.

## Data availability

All data used in this study have been obtained through numerical simulations that can be reproduced using the methods described in the paper and the codes publicly available at https://github.com/LValentinaGambuzza/Code-for-Reconstructing-higher-order-interactions-in-coupled-dynamical-systems.

## Code availability

All codes used in our work are based on the MATLAB commands described in the "Methods". They are publicly available in the following repository: https://github.com/LValentinaGambuzza/Code-for-Reconstructing-higher-order-interactions-in-coupled-dynamical-systems.

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

## Acknowledgements

This work was supported by the MUR PNRR Extended Partnership Initiative on Emerging Infectious Diseases (Project No. PE00000007, INF-ACT).

## Author contributions

G.R., V.L., L.V.G., and M.F. conceived the research and developed the theoretical framework for the analysis. F.M., G.R., and M.F. wrote the code. F.M. and M.F. performed the numerical simulations. F.M., A.C., and L.V.G. analyzed the data. All authors examined the results and wrote the manuscript.

## Competing interests

The authors declare no competing interests.
