## [Peer Review File · Nature Communications]

REVIEWER COMMENTS

Reviewer #1 (Remarks to the Author):

This paper studies the general problem of reconstruction of higher-order interactions from state vectors generated by a coupled ODE system.

Reconstruction is approached as the solution of N (potentially under-determined) systems of equations reminiscent of compressed sensing. The method is demonstrated on two problems; one has to do with community ecology and the other with oscillator dynamics on a social structure.

I liked the paper; it is well-written, short, and easy to read. The demonstration is compelling and presents a nice & direct solution to the stated problem. The problem itself is important to solve well. And as far as I can tell --- this is a fast-moving literature with deep roots-- -- this is the first time I've seen this approach to reconstruction done. So I see the potential for this paper being a nice "anchor point" for the literature going forward.

I'm not that it is currently sufficiently fleshed out to live up to this potential.

First, to my mind, a good anchor point cites its historical roots clearly. And while the paper does a good job of establishing itself in the context of the literature on higher-order networks, it skims on its relationship with other methodologies---and given that it is a methods paper, the latter seems important. Rewriting discretized coupled ODE as a system of equations to be

solved with LSE, possibly with regularization terms, is standard. I can't find an authoritative source quickly but see, for example, "Network Structure Reconstruction with Symmetry Constraint" by Hang et al., which I recently read and thus comes to mind. There are also evident similarities with compressed sensing and network reconstruction in general. I would encourage the authors to identify key references / clearly show how they build on these ideas.

Second, several critical methodological questions are essential to raise / address in a reference paper such as this.

1. Clarify and expand the discussion of identifiability. The identifiability of the structural sensor surely depends on more than the sampling rate (i.e., whether A is under or over-determined.) For example: can one "hack" an experiment and make tons of measurements in close succession to get good reconstruction by having $M/H > 1$, even if they all capture the same physics? Or: are there structural configurations (two different tensors A) that cannot be distinguished from Eq. (3) even if the equations are fully determined? What are the conditions?

2. What kind of solvers are used for OLS? Gradients-based? Or integer programming? (Or is weighted = continuous here?) More clarity on this and the domain of A seems necessary. The second experiment gives a bit more detail, but not the first.

3. Related to the above: Do we expect the problem to be "easy"? For binary tensors, I would assume not by reduction to other NP-Hard problems, while for continuous tensors, the MSE objective is convex and thus not a problem.

4. The method reconstructs the same k -dimensional interaction from the perspective of k nodes. Are these expected to agree? How should one handle consistency?

5. Reconstruction from perfect observations is usually trivial, up to the identifiability issues raised in my first point-- how does the method handle measurement errors/stochasticity? Is there anything else that can be said beyond the normal OLS results with i.i.d. normal noise? Are correlations important here, given that we're modeling relationships between units at multiple dimensions?

6. How does the method scale? The Φ matrix can become large. Are there tricks? Can one exploit sparsity? Small note: the manuscript does not state it, but I think it also assumes that the coupling strengths σ are known.

In sum, I see the potential for an outstanding paper but encourage the authors to flesh it out so that it may become *the* reference on higher-order network reconstruction.

Reviewer #2 (Remarks to the Author):

The manuscript "Reconstructing high-order interactions in coupled dynamical systems" by F. Malizia et al considers a very important and timely problem. It is the problem of network reconstruction from the time evolution of continuous variables describing a complex system. Authors are presenting an optimization-based approach able to infer the higher-order structural connectivity of the complex system. The method, rather general in principle, is applied to two specific applications mimicking (i) an ecosystem of bacteria and (ii) a set of interactions between and among chaotic oscillators.

The method is original, elegant and flexible, and the results obtained with the two presented examples are convincing. The manuscript is therefore a timely and important contribution to the growing literature of network reconstruction in the presence of higher-order interactions. It deserves publication in Nature Communications. The manuscript is well written and my only suggestion to the authors is to provide a brief statement in the discussion clarifying the computational limits of their method. In their examples, authors investigated higher-order structures involving up to 3 nodes. Can systems with hyperlinks or simplices with 4 or 5 nodes also be investigated? A brief statement about the computational limitations of the method would clarify the set of complex systems where the proposed method could be fruitfully applied.

Rosario Nunzio Mantegna

Revisions for Nature Communications manuscript NCOMMS-23-20846, version 1
Reconstructing higher-order interactions in coupled dynamical systems
by Federico Malizia, Alessandra Corso, Lucia Valentina Gambuzza, Giovanni Russo,
Vito Latora, Mattia Frasca

RESPONSES TO REVIEWER 1

This paper studies the general problem of reconstruction of higher-order interactions from state vectors generated by a coupled ODE system. Reconstruction is approached as the solution of N (potentially under-determined) systems of equations reminiscent of compressed sensing. The method is demonstrated on two problems; one has to do with community ecology and the other with oscillator dynamics on a social structure.

I liked the paper; it is well-written, short, and easy to read. The demonstration is compelling and presents a nice & direct solution to the stated problem. The problem itself is important to solve well. And as far as I can tell — this is a fast-moving literature with deep roots— — this is the first time I’ve seen this approach to reconstruction done. So I see the potential for this paper being a nice ”anchor point” for the literature going forward.

We thank the Referee for the time spent in reviewing our paper, for their valuable comments and for appreciation of our work.

I’m not that it is currently sufficiently fleshed out to live up to this potential. First, to my mind, a good anchor point cites its historical roots clearly. And while the paper does a good job of establishing itself in the context of the literature on higher-order networks, it skims on its relationship with other methodologies—and given that it is a methods paper, the latter seems important. Rewriting discretized coupled ODE as a system of equations to be solved with LSE, possibly with regularization terms, is standard. I can’t find an authoritative source quickly but see, for example, “Network Structure Reconstruction with Symmetry Constraint” by Hang et al., which I recently read and thus comes to mind. There are also evident similarities with compressed sensing and network reconstruction in general. I would encourage the authors to identify key references / clearly show how they build on these ideas.

The Referee is correct and thank them for this precious comment. Our method relies on the derivation of a set of algebraic equations for the parameters that appear in the ODE model under consideration, following a similar approach employed in addressing identification problems within other contexts [1–5]. The unknown terms in the algebraic equations are determined through least-square minimization, possibly applying regularization methods. Such unknown terms can either describe the ways in which the variables are combined together (namely the coefficients of the different linear and nonlinear terms effectively appearing in the equations of a single dynamical unit [1]), or the way in which variables from different units are effectively coupled (namely the weights of the pairwise interactions in a network of coupled dynamical units [2–5]). In the first case, the approach aims at deriving the governing equations of an unknown dynamical system from measurements of its state [1], whereas in the second case the functional form of the local dynamics is supposed to be known and the approach aims at reconstructing the interactions among the units [2, 3, 5]. In both cases, the problem is sparse, as various dynamical systems can be modeled by using a low number of nonlinear terms from the large dictionary of possible nonlinear functions, and the interactions among the units of a complex system are often mapped into a sparse network. This enables the use of optimization problems based on compressive sensing, such as the lasso method, to solve this type of problems. While previous works [2, 3, 5] have addressed the reconstruction problem of pairwise networks, the method we propose in our manuscript deals with the more general case of structures with interactions of any order.

We have included these considerations in the revised version of our paper (introductory section).

Second, several critical methodological questions are essential to raise / address in a reference paper such as this.

1. Clarify and expand the discussion of identifiability. The identifiability of the structural sensor surely depends on more than the sampling rate (i.e., whether A is under or over-determined.) For example: can one "hack" an experiment and make tons of measurements in close succession to get good reconstruction by having $M/H > 1$, even if they all capture the same physics? Or: are there structural configurations (two different tensors A) that cannot be distinguished from Eq. (3) even if the equations are fully determined? What are the conditions?

We thank the referee for raising this issue. As any other problem of identification, the identifiability of the structural tensor mainly depends on the collected data, and is difficult, if not impossible, to establish a-priori conditions for the identifiability. In our work, we formulate the identification task in terms of different optimization problems, depending on the type of structure and also on the features of the data available. For instance, in the case of unweighted networks, we propose to use three different methods, namely ordinary least squares (OLS), signal Lasso and Non-negative least squares (NNLS) methods. The latter two are based on a regularization of the problem, and provide a unique/optimal solution even if $M < H$. If the M observations are able to sample M different states of system dynamics, thus producing new information at each new measurement, then all techniques will converge to the unique solution that perfectly identifies the system as soon as $M \geq H$. However, if it happens that, when we increase the number of measurements M we do not get extra information (because the new measurements all capture the same physics), this means that we are adding new equations to our system that are linear combination of the equations we already have. Hence, even if $M > H$ our system may be still underdetermined, and the regularization methods have to be used, providing the optimal solution given the insufficient data. About the last question of this point, if the equations are fully determined, i.e. if matrices Φ_i are full rank, then the system solutions (i.e. the tensors \mathcal{A}_i) are uniquely determined. Hence, in this case, there cannot be two structural configurations that yield the same signals.

In the revised version of our manuscript, we have now discussed the problem of identifiability at the end of the section entitled "Reconstructing pairwise and higher-order interactions".

2. What kind of solvers are used for OLS? Gradients-based? Or integer programming? (Or is weighted = continuous here?) More clarity on this and the domain of A seems necessary. The second experiment gives a bit more detail, but not the first.

We thank the Referee for this observation. Indeed in the first experiment, namely in the case of a weighted higher-order structure reported in section "Lotka-Volterra dynamics on weighted directed hypergraphs", we assume that the weights are continuous. We have now added a comment to this section to specify that the weights are real numbers.

The solver employed here in our study for OLS is based on the complete orthogonal decomposition (see [6]). We have added a section in the methods, where we now clarify this point as well as the MATLAB function used for OLS and for the other two methods adopted in the paper, NNLS and SL.

3. -COMPLEXITY- Related to the above: Do we expect the problem to be "easy"? For binary tensors, I would assume not by reduction to other NP-Hard problems, while for continuous tensors, the MSE objective is convex and thus not a problem.

The case of OLS is standard for structures with continuous weights while, as correctly noted by the Referee, the case of unweighted structures may be more complex because it involves a combinatorics minimization. In the latter case, we decided to adopt penalization methods that reduce discrete optimization to a continuous one, thus simplifying the search for the optimal solution. This is now explained in more detail in the section "Coupled Rössler oscillators on undirected simplicial complexes".

4. -DONE- The method reconstructs the same k -dimensional interaction from the perspective of k nodes. Are these expected to agree? How should one handle consistency?

We thank the Referee for pointing this out. When dealing with directed interactions, as in the first study case presented in our manuscript, there are no consistency problems since the various systems in Eq. (3) are independent.

There might be however cases when we have an a-priori knowledge that the interactions to reconstruct are symmetric. i.e. that the underlying structure to reconstruct is undirected. This is for

instance what happens in our second case study, namely in the system of coupled Rössler oscillators in undirected unweighted simplicial complexes. In this example, we have to reconstruct an undirected structure, namely the simplicial complex of the Karate club, from the time series of node activities. In such a case the systems in Eq. (3), $i = 1, \dots, N$, are no longer independent. However, we can still treat the problem as if the underlying structure was directed, meaning that, e.g. for a given pair of connected nodes i and j we consider both terms $a_{ij}^{(1)}$ and $a_{ji}^{(1)}$ in the adjacency matrix as independent unknowns to be determined. Analogously, for a 3-node simplex (i, j, k) , we have six independent unknowns $a_{ijk}^{(2)}$ in the tensor, one for each permutation of the three indices i, j and k . When M is large enough, so that the error is zero, the obtained tensors will automatically result symmetric, i.e. $a_{ij}^{(1)} = a_{ji}^{(1)}$ for the first order and $a_{ijk}^{(2)} = a_{ikj}^{(2)} = a_{jik}^{(2)} = a_{jki}^{(2)} = a_{kij}^{(2)} = a_{kji}^{(2)}$ for the second order. A discussion about this point is added in the new version of our manuscript (section results). In general, in order to avoid inconsistency one should minimize the sum of the discrepancies at each node, as indicated in Eq. (6). Alternatively, in order to simplify the application of the least square techniques, one could treat the structures as directed, and add a penalization on the lack of symmetry of the tensors. We mentioned this possibility in the discussion section.

In other situations, for example in presence of noise when strict consistency might be violated by straightforward application of a directed framework, the symmetry of the coefficients may be imposed by other techniques, for example using penalization on the lack of symmetry. A remark about this, is mentioned in the discussion.

5. -NOISE- Reconstruction from perfect observations is usually trivial, up to the identifiability issues raised in my first point— how does the method handle measurement errors/stochasticity? Is there anything else that can be said beyond the normal OLS results with i.i.d. normal noise? Are correlations important here, given that we're modeling relationships between units at multiple dimensions?

We thank the Referee for this observation that was a stimulus for broadening the problem dealt in our paper. Specifically, we have now considered two cases: 1. the derivatives of the time evolution of the state variables are known; 2. the derivatives are not available and need to be estimated numerically, such as through finite difference approximation. The second case, clearly, introduces a numerical error (the estimation error) which can be thought as a noise on the input signal of the method. This extended framework is much more general and relaxes the assumptions considered in the first version of our manuscript (that is, a perfect knowledge of the input signal of the method). In the new version of the paper, we have considered several approximation methods for the derivatives and show how the order of the approximation impacts the results. As a general conclusion, we have found that reconstruction is still possible, provided that a larger sample size M is available.

The manuscript has been now fully reorganized to discuss the theory of this more general framework and to include new simulation results. Consequently, Figs. 1 and 2 have been redrawn to illustrate the new findings. We retain this a very fundamental step that we overlooked in the first version of the manuscript and that was important to investigate in this new version.

Finally, we note that, since the reconstruction method proposed in this paper is based on the numerical approximation of the derivatives from the data, in the presence of noisy signals the method cannot be directly applied as is. In such a case one can resort to the more traditional approach based on the minimization of the discrepancy between the noisy measured trajectories and the computed ones that depend on the hypergraph parameters. However this technique leads to a strongly non-linear least square problem, which may be computationally very expensive. An alternative approach consists in application of our method after a suitable pre-processing of the measured data. The latter approach is currently under investigation. We have added a remark about this last point in the section "discussion".

6. - LARGE NETWORKS - How does the method scale? The Phi matrix can become large. Are there tricks? Can one exploit sparsity? Small note: the manuscript does not state it, but I think it also assumes that the coupling strengths σ are known.

In the current paper we have considered relatively small systems, in which there is no *a priori* knowledge of their structure. For large systems it is not feasible to consider all possible interactions, since the number of unknowns grows as N^D , where D is the maximum order of interaction we consider, making the problem practically intractable. In the case some basic information is available, for example if one knows the graph describing the pairwise backbone of the system, then one could limit, for

instance, the search of higher-order interactions of dimension 2 among the triangles of the graph. These considerations have been now included in the section "discussion" of the manuscript.

In sum, I see the potential for an outstanding paper but encourage the authors to flesh it out so that it may become *the* reference on higher-order network reconstruction.

We thank once again the Referee for the positive evaluation of the potential of our paper and we are confident that the new revised version of our manuscript has been significantly improved thanks to all comments and suggestions of the Referees.

RESPONSES TO REVIEW 2

The manuscript “Reconstructing high-order interactions in coupled dynamical systems” by F. Malizia et al considers a very important and timely problem. It is the problem of network reconstruction from the time evolution of continuous variables describing a complex system. Authors are presenting an optimization-based approach able to infer the higher-order structural connectivity of the complex system. The method, rather general in principle, is applied to two specific applications mimicking (i) an ecosystem of bacteria and (ii) a set of interactions between and among chaotic oscillators.

First of all, we would like to thank the Referee for the time spent in reviewing our paper, for his valuable comments and for appreciation of our work.

The method is original, elegant and flexible, and the results obtained with the two presented examples are convincing. The manuscript is therefore a timely and important contribution to the growing literature of network reconstruction in the presence of higher-order interactions. It deserves publication in Nature Communications. The manuscript is well written and my only suggestion to the authors is to provide a brief statement in the discussion clarifying the computational limits of their method. In their examples, authors investigated higher-order structures involving up to 3 nodes. Can systems with hyperlinks or simplices with 4 or 5 nodes also be investigated? A brief statement about the computational limitations of the method would clarify the set of complex systems where the proposed method could be fruitfully applied.

We thank the Referee for this interesting consideration. As we have now clarified (in the section “discussion”) the method is presented only for multi-body interactions of order 3, but is fully general and can be applied to any order of interaction. In addition, we have now included a brief discussion on the computational limits of our approach (in the section “discussion”). We have clarified that for large systems it is not feasible to consider all possible interactions, since the number of unknowns grows as N^D , where D is the maximum order of interaction we consider, making the problem practically intractable. In the case some basic information is available, for example if one knows the graph describing the pairwise backbone of the system, then one could limit, for instance, the search of higher-order interactions of dimension 2 among the triangles of the graph.

-
- [1] S. L. Brunton, J. L. Proctor, and J. N. Kutz, Proceedings of the national academy of sciences **113**, 3932 (2016).
 - [2] W.-X. Wang, Y.-C. Lai, C. Grebogi, and J. Ye, Physical Review X **1**, 021021 (2011).
 - [3] X. Han, Z. Shen, W.-X. Wang, and Z. Di, Physical Review Letters **114**, 028701 (2015).
 - [4] Z. Hang, P. Dai, S. Jia, and Z. Yu, Chaos, Solitons & Fractals **139**, 110287 (2020).
 - [5] L. Shi, C. Shen, L. Jin, Q. Shi, Z. Wang, and S. Boccaletti, Physical Review Research **3**, 043210 (2021).
 - [6] G. H. Golub and C. F. Van Loan, *Matrix computations* (JHU press, 2013).

REVIEWERS' COMMENTS

Reviewer #1 (Remarks to the Author):

I still very much like the manuscript and recommend it for publication.

The authors have improved it on all fronts.

****Literature Review:**** I particularly appreciate the addition of Brunton et al. for its relevance.

****Identifiability:**** There is not a lot of new discussion, but that is OK. The full rank condition is a nice and relevant addition.

****Methods**:** The new methods section adds much-needed clarity. It's a solid improvement.

****Technical Specifics**:** The clarifications on the solution process and the nature of weights respond directly to my initial concerns. While the additional details on the OLS solver were more than I expected, they've made the methodology clearer. The authors should balance detail with readability according to their best judgment.

****Future Directions**:** The ongoing work on noisy data is promising. I'm assuming some sort of smoothing of the trajectories will be necessary? Exciting work. I'm fine with not solving it in this landmark paper.

****Scalability**:** Acknowledging the scalability challenges is honest and highlights a broader issue in the field. It will be an important area for future research. Perhaps NN Descent methods in the spirit of <https://arxiv.org/abs/2401.01404>, but deployed with hypergraphs.

****Derivative Reconstruction**:** The section on derivatives reconstruction is a strong addition and showcases the practical application of the method from time series data. The particular setup might leave information on the table since the coupling functions are assumed to be known, meaning that one can calculate the derivatives conditioned on the tensor and a coordinate in the phase space.